# A Family Case of Congenital Myasthenic Syndrome-22 Induced by Different Combinations of Molecular Causes in Siblings

**DOI:** 10.3390/genes11070821

**Published:** 2020-07-19

**Authors:** Olga Shchagina, Ludmila Bessonova, Igor Bychkov, Tatiana Beskorovainaya, Aleksander Poliakov

**Affiliations:** Research Centre for Medical Genetics 1 Moskvorechie St., 115522 Moscow, Russia; bessonovala@yandex.ru (L.B.); bychkov.nbo@gmail.com (I.B.); t-kovalevskaya@yandex.ru (T.B.); apol@dnalab.ru (A.P.)

**Keywords:** CMS22, PREPL, uniparental disomy, splice site, spliceogenic

## Abstract

Congenital myasthenic syndrome-22 (CMS22, OMIM 616224) is a very rare recessive hereditary disorder. At the moment, ten CMS22 patients are described, with the disorder caused by nine different Loss-*of*-Function mutations and 14 gross deletions in the *PREPL* gene. The materials for our study were DNA samples of five family members: two patients with myasthenia, their healthy sibling and parents. Clinical exome analysis was carried out for one patient, then the whole family was checked for target variants with Sanger sequencing, quantitative multiplex ligation-dependent probe amplification, and chromosome 2 microsatellite markers study. To determine the functional significance of the splicing variant, we applied the minigene assay. The cause of the proband’s disorder is a compound heterozygous state of two previously non-described pathogenic *PREPL* variants: a c.1528C>T (p.(Arg510Ter)) nonsense mutation and a c.2094G>T pseudo-missense variant, which, simultaneously with a p.(Lys698Asn) amino acid substitution, affects splicing, leading to exon 14 skipping in mRNA. The second patient’s disorder was caused by a homozygous nonsense c.1528C>T (p.(Arg510Ter)) mutation due to maternal uniparental disomy (UPD) of chromosome 2. In this study, we describe a unique case, in which two siblings with a rare disorder have different pathologic genotypes.

## 1. Introduction

Congenital myasthenic syndrome-22 (CMS22, OMIM 616224) is a very rare recessive hereditary disorder. Its characteristic traits are severe neonatal hypotonia, muscular weakness and feeding difficulties [1,2]. Hypotonia and feeding problems become less noticeable during the first year of life; however, ptosis, nasal dysarthria, facial weakness, and proximal muscular weakness remain present. In childhood, patients develop hyperphagia with tendencies towards obesity. This type of myasthenia is caused by homozygous and compound heterozygous Loss-*of*-Function (LoF) mutations in the *PREPL* gene. An allelic variant of this disorder is hypotonia-cystinuria syndrome (HCS, OMIM # 606407), with cystinuria added to the symptoms listed above. HCS, unlike CMS22, is caused by gross deletions including two adjacent genes on the chromosome 2p21: *SLC3A1*, which codes the heavy-chain subunit of the cystine and dibasic amino acid transporter (OMIM * 104614), and *PREPL*, which may affect function of the clathrin-associated adaptor protein-1 [3] essential for normal trafficking of the vesicular ACh transporter between the synaptic vesicle membrane and the cytosol [4] (OMIM * 609557). Homozygous and heterozygous *SLC3A1* mutations lead to cystinuria (OMIM # 220100).

At the moment, ten CMS22 patients are described, with the disorder caused by nine different LoF mutations and 14 gross deletions in the *PREPL* gene [2,5,6,7,8]. In one instance, the cause was maternal uniparental disomy of chromosome 2, which carries the frameshift mutation in the *PREPL* gene [8].

In this study, we report a family case of congenital myasthenic syndrome-22. Its distinctive feature is different genetic causes of the disorder in two siblings. One of them has two previously non-described *PREPL* pathogenic variants: nonsense c.1528C>T (p.(Arg510Ter)) and missense/splice site variant c.2094G>T (p.(Lys698Asn)) in a compound-heterozygous state; the other has maternal uniparental disomy of chromosome 2, which carries a c.1528C>T (p.(Arg510Ter)) variant.

## 2. Materials and Methods

DNA was extracted from whole blood samples using a Wizard^®^ Genomic DNA Purification Kit (Promega, Madison, WI, USA) according to the manufacturer’s protocol.

The proband’s DNA was analysed by «GENOTEK» company (Moscow, Russia) using paired-end reading (2 × 75 bp) on an IlluminaNextSeq 500 sequencer. The probe was prepared using a Genotek Clinical Exome Kit (Illumina Inc., San Diego, CA, USA). The detected variants were named according to nomenclature presented on the http://varnomen.hgvs.org/recommendations/DNA website (version 2.15.11).

Sequencing results were analysed using a standard Illumina automatised algorithm for data analysis, presented on the https://basespace.illumina.com website. Average coverage for this sample was 51.8×, coverage width (10×)—95.62%.

Automatic Sanger sequencing was carried out using ABIPrism 3100xl Genetic Analyzer (Applied Biosystems, Foster City, CA, USA) according to the manufacturer’s protocol. Primer sequences were chosen according to the NM_001171603.1 reference sequence.

To determine functional significance of the c.2094G>T (p.(Lys698Asn)) variant, we applied the minigene assay as previously described [9]. Exon-of-interest with about 300 bp of flanking intronic sequences was cloned into pSpl3-Flu2 vector. Wild type (WT) and c.2094G>T reporter plasmids were transfected into HEK293T cells with the CaPO4 method. After 48 h, the RNA was extracted and reverse transcribed.

Microsatellite markers from the «AmpFlSTR Identifiler Direct PCR Amplification Kit» (Applied Biosystems, Foster City, CA, United States), and *D2S119*, *D2S2174*, and *D2S2294* markers flanking the *PREPL* gene, were used for genotyping.

Quantitative analysis was carried out using the SALSA MLPA Probemix P426 Cystinuria Kit (MRC-Holland).

Sequencing results for 1036 exomes (2072 chromosomes, sequenced on IlluminaNextSeq 500 using the IlluminaTruSeq^®^ ExomeKit for probe preparation) of non-related Russian patients with various hereditary pathologies were used as control.

All procedures performed in studies involving human participants were in accordance with the ethical standards of the WMA declaration of the Helsinki “Ethical principles for medical research involving human subjects” (https://www.wma.net/policies-post/wma-declaration-of-helsinki-ethical-principles-for-medical-research-involving-human-subjects/) protocol number 5/8, dated 12 November 2018.

This article does not contain any studies with animals performed by any of the authors.

Informed consent was obtained for genetic examination and publication with anonymity from all patients or their legal representatives.

## 3. Results

### 3.1. Clinical Information

Two affected male siblings born in 2010 (Patient 1) and 2016 (Patient 2) are the second and third children in the family. Their older brother, born in 2005, is healthy. Their parents are Russian, kinship denied: the father was born in 1981, the mother in 1983. Patient 1 was born via Caesarean section at 39–40 weeks, weighed 3210 g, length 50 cm, 10/10 APGAR score. Patient 2 was born at 39–40 weeks of gestation, weighed 3270 g, length 50 cm, 8/8 APGAR score. On the third day of life, both patients developed noticeable hypotonia, apathy and suck reflex suppression. They were transferred to an intensive therapy ward, where a nasogastric tube was installed. They were fed through it up to the age of 11 months. Early psychomotor development: Patient 1—hold head since 5 months, turn since 6 months, sit since 8 months, walk since 1 year 5 months; Patient 2—hold head since 3 months, turn since 6 months, sit since 8 months, walk since 1 year 5 months. Their first words were uttered at the age of 1 year. Self-service skills developed appropriately to age.

Patient 1 (born in 2010) was examined at the age of 8. He attended elementary school and had learning difficulties. His height was 121 cm and his weight was 30.4 kg (growth retardation, excess weight). Patient 2 (born in 2016) was examined at the age of 2 years and 6 months, his height was 84 cm and his weight was 9.8 kg.

Both patients had mild bilateral ptosis, nasal voice, dysarthria, dysphonia. They couldn’t keep their saliva in their mouths. Neurological examination showed reduced pharyngeal and palatal reflexes, hypotonia, inability to hold hands above head for a long time. Both patients were able to walk without support, had a waddling gait on a wide base with mild rotation of the feet inwards and on the medial part of the feet. Deep tendon reflexes in the extremities were retained and symmetric.

Nerve conduction studies of peroneal, tibial and median nerves of both sides revealed normal ranges of compound muscle action potential amplitude and terminal latencies; conduction velocities were also normal. The amplitude of sensory nerve action potential of both sural and median nerves was normal. Standard 3 Hz repetitive stimulation of the right median nerve (abductor pollicis brevis) revealed no the decrement increasing, as well as repetitive stimulation of the right facial nerve (posterior belly of digastric muscle) for Patient 2. Brain MRI did not reveal any abnormalities. Biochemical blood analysis showed normal creatine kinase, alanineaminotransferase (ALT), aspartateaminotransferase (AST), Lactatedehydrogenase (LDH), and lactate levels. Complex urine amino acid analysis did not reveal any abnormalities. Hormone analysis showed decreased IGF-1 levels (Patient 1—61 ng/μL, Patient 2—52 ng/μL, reference 85—316 ng/μL), but normal levels of TSH, free T4, cortisol and insulin.

### 3.2. Molecular Analysis

Massive parallel sequencing for Patient 2 revealed two previously non-described *PREPL* pathogenic variants: c.1528C>T leading to premature stop codon formation p.(Arg510Ter) in exon 10 and missense variant c.2094G>T (p.(Lys698Asn)) in exon 14. Sanger sequencing of the target fragments for the parents showed a trans configuration of these variants in the patient. The results of Sanger sequencing of family members are presented in Appendix A. The c.1528C>T variant (2-44556077-G-A) was present on one chromosome in the African population out of 31 394 total in the gnomAD database (allelic frequency is 0.000032), c.2094G>T (2-44548966-C-A)—on four chromosomes in the European (non-Finnish) population out of 251 208 (allelic frequency is 0.000016), never in the homozygous state. Neither of the variants was detected on 2072 chromosomes (1036 exomes) of non-related Russian patients with various hereditary disorders.

There are no missense variants decsribed as pathogenic in the *PREPL* gene. However, the c.2094G>T variant is located in the donor splice site of exon 14 and, as predicted by SpliceAI, significantly decreases its strength (delta score for donor loss 0.92 with the threshold of > 0.2 for the variant to be significant) [10]. Functional analysis using the minigene assay was performed to validate the effect of the c.2094G>T variant on splicing. The results are presented in Figure 1.

As shown on Figure 1, minigene assay demonstrates that the c.2094G>T variant causes the complete absence of the full-length transcript isoform with exon 14. Skipping of the exon 14 corresponds to r.2021_2094del (p.Gly674Aspfs*6) and leads to frameshift with formation of the premature stop codon in the last exon of the gene. Therefore, the mRNA avoids the nonsense mediated decay and the truncated protein lacking 55 C-terminal aminoacids is synthesized. Nevertheless, this deletion involves the catalytic residue Hys690, which is critical for protein function [11].

A DNA analysis for Patient 1 showed unexpected results: variant c.2094G>T (p.(Lys698Asn)) was not found, variant c.1528C>T (p.(Arg510Ter)) was detected, probably in a homozygous state. To determine the variant’s zigosity, quantitative multiplex ligation-dependent probe amplification (MLPA) was carried out. The MLPA kit included probes for exons 1, 2, 4–7, 9–13, 15 of the *PREPL* gene and exons 1–10 of the *SLC3A1* gene. The analysis showed two copies of the tested fragments, including *PREPL* exon 10, where the c.1528C>T (p.(Arg510Ter)) nonsense variant is located, in all family members. Results of quantitative analysis using SALSA MLPA Probemix P426 Cystinuria are presented in Appendix A. This indicates the absence of *PREPL* gene deletion in Patient 1 as well as other family members. Genotyping with markers from the AmpFlSTR Identifiler kit showed the absence of a paternal allele at marker *D2S1338* in Patient 1; both alleles have maternal origin. Genotyping results of chromosome 2 markers are presented in Appendix A. To confirm the maternal uniparental disomy (UPD) in Patient 1, family members were genotyped using the *PREPL* flanking markers *D2S119, D2S2174*, and *D2S2294*. The results are presented in Figure 2.

As shown in Figure 2, the chromosome 2 haplotypes confirm maternal UPD in the *PREPL* region in Patient 1. Moreover, the healthy sibling and Patient 2 each inherited a different chromosome 2 from the father.

## 4. Discussion

To date, there are nine minor mutations and 14 gross deletions described in the *PREPL* gene. All the mutations are LoF variants: three leading to a premature terminating codon formation, five leading to a frameshift, and two canonical splice site mutations. Two variants detected in the examined family are also LoF: c.1528C>T leads to a premature terminating codon p.(Arg510Ter) formation, and the missense c.2094G>T (p.(Lys698Asn)) variant for which we present the evidence, that it is the spliceogenic LoF variant that leads to the synthesis of the truncated PREPL protein missing part of the catalytic site.

One of the affected siblings had maternal uniparental disomy of chromosome 2. There are three mechanisms described of uniparental disomy leading to disorders: imprinting, mosaic aneuploidy or homozygosity for a recessive pathogenic variant [12]. Maternal and paternal uniparental disomy of chromosome 2 were described in humans with a normal phenotype [13,14,15]. Therefore, the cause of the disorder in Patient 1 is a homozygous pathogenic c.1528C>T (p.(Arg510Ter)) variant originated from maternal uniparental disomy.

Clinical findings were practically identical in both affected siblings and perfectly matched the description of congenital myasthenic syndrome-22. All clinical differences can be explained by the patients’ age at the moment of examination: e.g., the younger patient did not have excess weight or growth retardation. However, his hormonal analysis also showed low levels of IGF-1, which is characteristic for this disorder, as PREPL is involved in growth hormone secretion. Because none of the detected variants affect the *SLC3A1* gene, which is located close to *PREPL,* neither of our patients had cystinuria, which is described in some CMS22 cases as a result of the compound heterozygous pathogenic *PREPL* variant and gross deletion of region 2p21 affecting both of these genes.

In this study, we describe a rare case of two siblings having different pathologic genotypes. In one patient, the disorder is caused by two compound heterozygous pathogenic *PREPL* variants: a nonsense c.1528C>T (p.(Arg510Ter)) variant and a pseudo-missense c.2094G>T variant, which we reclassified as a spliceogenic LoF variant.

In the other patient, the disorder is caused by a homozygous nonsense c.1528C>T (p.(Arg510Ter)) variant due to maternal UPD of chromosome 2.

## Figures and Tables

**Figure 1 genes-11-00821-f001:**
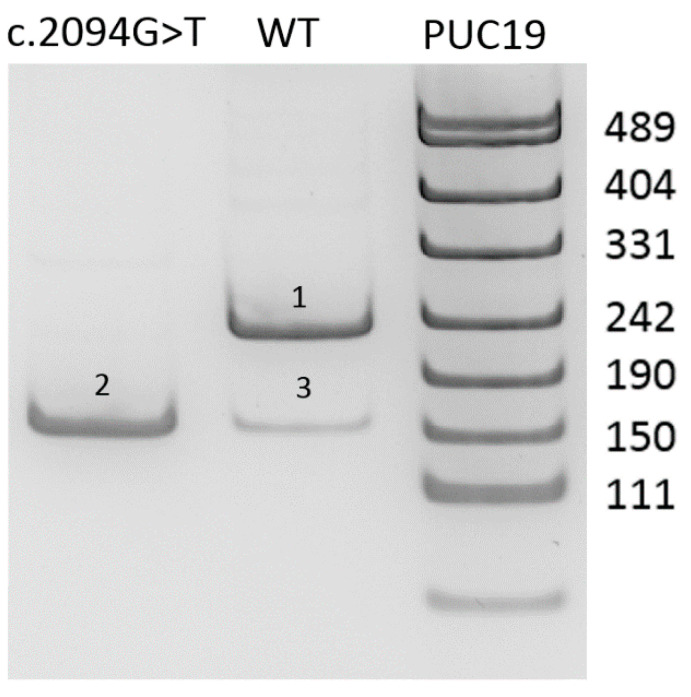
Results of minigene assay. Visualisation of PCR products from amplification of minigene-specific cDNA. 1—Wild type (WT) transcript isoform. 2—transcript isoform with *PREPL* exon 14 skipping, caused by c.2094G>T variant. 3—minor event of exon skipping in WT minigene. As can be seen from our experiment, alternative splicing with the formation of a short transcript is also present in the wild type. This mRNA isoform probably avoids the nonsense mediated decay and is therefore present in small amounts in normal cells as well.

**Figure 2 genes-11-00821-f002:**
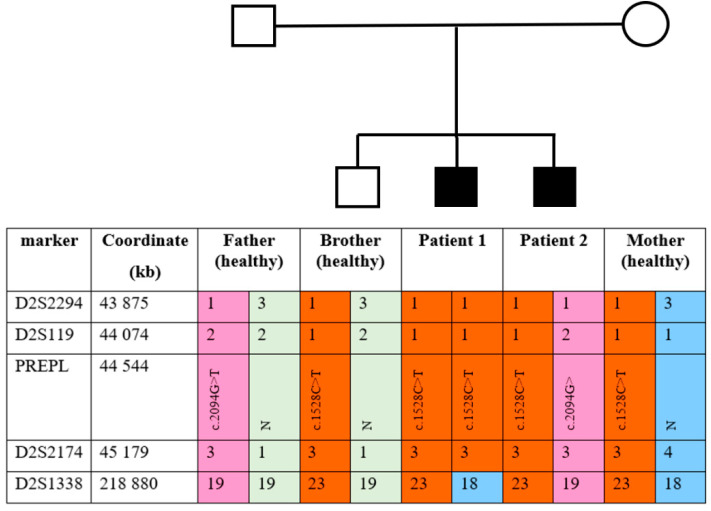
The family’s haplotypes at chromosome 2 markers.

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
