# Peer review of "A Family Case of Congenital Myasthenic Syndrome-22 Induced by Different Combinations of Molecular Causes in Siblings"

_genes, 2020, doi:10.3390/genes11070821_

Round 1
Reviewer 1 Report
The Authors describe a congenital myasthenic syndrome due to PREPL mutations, in two siblings. There are several points to consider
- In the description of the cases (Clinical informations) you write "Both patients are able to walk without support, have waddling
gait with elements of ataxia. What do you mean by minimal elements of ataxia? Please explain - Please instead of the term EMNG, use conventional terms: Electromyography (EMG) and nerve conduction velocity studies (NCV studies). "Examination revealed signs of primary myopathic process...": do you mean that motor unit potentials amplitude (MUPs) was small and their duration was reduced ? How was the maximum effort pattern ? Was it also compatible with a myopathic process ? Please explain better. Replace the term "amplitude of the M-response" with the term amplitude of compound muscle action potential (cMAP). The sentence "Rhytmic stimulation of N. medianus... , should be reworded: Repetitive stimulation of the median nerve (left ? right ?) ..... About the muscles used for recordings data (m.abd.pol.br.) please write in full and add posterior to digastric
- Has a therapeutic attempt with pyridostigmine been made ? Since positive effects have been reported in the few cases reported. Please add a comment about.
- Have you any data on follow-up ? Please add a comment
Author Response
Dear reviewer, thank you for your valuable remarks.
Point 1: In the description of the cases (Clinical informations) you write "Both patients are able to walk without support, have waddling
gait with elements of ataxia. What do you mean by minimal elements of ataxia? Please explain
Response 1: Gait on a wide base with mild rotation of the feet inwards and on medial part of the feet in both patients. The description of the gait as with ataxia is changed in the report and a detailed description of the gait features is added.
Point 2: Please instead of the term EMNG, use conventional terms: Electromyography (EMG) and nerve conduction velocity studies (NCV studies). "Examination revealed signs of primary myopathic process...": do you mean that motor unit potentials amplitude (MUPs) was small and their duration was reduced ? How was the maximum effort pattern ? Was it also compatible with a myopathic process ? Please explain better. Replace the term "amplitude of the M-response" with the term amplitude of compound muscle action potential (cMAP). The sentence "Rhytmic stimulation of N. medianus... , should be reworded: Repetitive stimulation of the median nerve (left ? right ?) ..... About the muscles used for recordings data (m.abd.pol.br.) please write in full and add posterior to digastric
Response 2: Thanks to you, we have engaged a specialist in electrophysiology to review the NCS and EMG data of both patients. Unfortunately, needle EMG was not performed in our patients. We have tried to present all the data in the text in an acceptable form.
Point 3: Has a therapeutic attempt with pyridostigmine been made ? Since positive effects have been reported in the few cases reported. Please add a comment about.
Response 3: Both patients were tested by using the intramuscular injection of neostigmine methylsulfate. There was no clinical positive effect, and it was decided not to prescribe pyridostigmine bromide.
Point 4: Have you any data on follow-up ? Please add a comment
Response 4: The family lives far from Moscow in the city Chita in the Transbaikal region, so we are not able to follow-up patients by ourselves. We know from Skype that there are no obvious changes in the condition of both sibs today. The older patient continues to study at an elementary school, the younger one is preparing to come to school.
Reviewer 2 Report
The manuscript is very innovative, well written and very detailed with in-depth experiments to demonstrate maternal uniparental disomy in Patient 1.
I would only suggest that the coverage could be completed from 65.62% to 100% with an automatic sanger sequencing to analyse the missing regions and exclude other possible pathological variants.
I would suggest minor revisions:
- line 139: remove "probably" because Figure S1 for the variant c.1528C> T shows that the patient has only one thymine in position 1528;
- line 163: specify why band n.3 appears in the WT minigene;
- Line 215: remove the character "
- Lines 241-243: these lines are not indented
- Figure S2: specify in the footnotes the meaning of 953-954-956 samples
Author Response
Dear reviewer, thank you for your valuable remarks.
Point 1: I would only suggest that the coverage could be completed from 65.62% to 100% with an automatic sanger sequencing to analyse the missing regions and exclude other possible pathological variants.
Response 1: We have repeatedly reviewed the "raw" NGS data and asked bioinformatics specialists from another organization to do this. The entire sequence of the PREPL gene is covered very well. And we do not see the point in reworking the exome, because these pathogenic variants fully correspond to the clinical phenotype of our patients. In this situation we did not see the need to examine the entire gene by Sanger sequencing.
Point 2: I would suggest minor revisions:
- line 139: remove "probably" because Figure S1 for the variant c.1528C> T shows that the patient has only one thymine in position 1528;
- line 163: specify why band n.3 appears in the WT minigene;
- Line 215: remove the character "
- Lines 241-243: these lines are not indented
- Figure S2: specify in the footnotes the meaning of 953-954-956 samples
Response 2: We have made changes in accordance with your comments.
line 140: remove "probably"
line 164: added to the text: As can be seen from our experiment, alternative splicing with the formation of a short transcript is also present in the wild type. This mRNA isoform probably avoids the nonsense mediated decay and is therefore present in small amounts in normal cells as well
Line 215: The character " is removed.
Lines 241-243: these lines are not indented - The text is formatted
Figure S2: specify in the footnotes the meaning of 953-954-956 samples - We added to the text an explanation that these are control samples of healthy adults